# Hesitancy towards COVID-19 vaccination: The role of personality traits, anti-vaccine attitudes and illness perception

Eric Nanteer-Oteng[1]*, Irene A. Kretchy[2], Deborah Odum Nanteer[2‡], James-Paul Kretchy[3‡], Joseph Osafo[1‡]

**1** Department of Psychology, University of Ghana, Legon, Accra, Ghana, **2** Department of Pharmacy Practice and Clinical Pharmacy, School of Pharmacy, University of Ghana, Legon, Accra, Ghana, **3** Public Health Unit, School of Medicine and Health Sciences, Central University, Miotso, Accra, Ghana

☯ These authors contributed equally to this work.
‡ DON, JPK and JO also contributed equally to this work.
* enanteer-oteng@st.ug.edu.gh

## Abstract

There is an increased need for COVID-19 vaccination since the world is gradually returning to normal. Current evidence supports vaccination activity more towards viral suppression than COVID-19 prevention. This has led to divergent views regarding vaccination which may influence anti-vaccine attitudes and vaccine hesitancy. The study examined the role of personality traits, anti-vaccine attitudes and illness perceptions on vaccine hesitancy. The study was a cross-sectional survey using snowball and convenience sampling to recruit 492 participants via social media platforms. Multivariate analysis of variance and regression analysis were used to test the hypotheses. The study found that some facets of illness perception (identity, concern, emotional representation and treatment control), extraversion, experience with COVID-19 and anti-vaccine attitudes (mistrust, profiteering, worries about unforeseen effects of vaccine) predicted vaccine hesitancy. The outcomes from this study have implications for achieving public health goals and developing strategies for reaching optimal vaccination targets and attaining herd immunity. Health-promoting programs need to be intensified and could include psychosocial perspectives on vaccine hesitancy so that specific target groups can be reached to be vaccinated.

## Introduction

Coronavirus disease 2019 (COVID-19) has quickly spread over the world since 2019. There have been over 479 million confirmed cases and an estimation of 6 million deaths as of 27th March 2022 with over 6 million deaths reported globally [1]. With many countries getting their citizens to actively participate in getting vaccinated against COVID-19 infection, Ghana is also pulling its weight with about 5,070,306 representing 16% of the total population being vaccinated as of 21st March 2022 according to the Ghana Health Service [2]. Little is known about the Ghanaian people's inclination to get vaccinated. People may decline to be vaccinated

**Funding:** The authors received no specific funding for this work.

**Competing interests:** The authors have declared that no competing interests exist.

for medical, religious, ideological, or situational factors. As vaccine refusal for nonmedical reasons becomes more prevalent in Ghana [3], there is an urgent need to develop motivated reasoning perspectives to overcome vaccination concerns and fallacies, and thus build effective interventions to boost vaccination rates.

Vaccines provide us with protection from diseases, consequently, the COVID-19 vaccines were made to curtail the global transmission of the virus. Nevertheless, the vaccine has been met with reluctance, anxiety, and safety worries along with conspiracy theories against vaccinations, which seriously undermine acceptability and readiness to receive the vaccine. The degree of fear and apprehension expressed by the public concerning vaccine acceptance and uptake is partially due to a lack of adequate information about the concepts behind vaccine development from the various phases of the vaccine trials that consider immunology, toxicity, and effectiveness concerns. Furthermore, the speed with which the COVID-19 vaccines were developed has particularly been identified as a significant contributor to the hesitation associated with vaccination adoption in many regions of the world, including Ghana, giving impetus to this study [4].

With the current data supporting the fact that COVID-19 vaccination does not prevent one from contracting the virus [5] but only suppresses potential symptoms that come with contracting it [6], this has led to many discourses of divergent views and attitudes towards vaccination. The divergent views may have some influence on anti-vaccine attitudes and vaccine hesitancy. Vaccine hesitation has been described as individuals' general unwillingness to be vaccinated although there is evidence of its safety and efficacy [7].

A study by Huyn et al. [8] showed that individuals who held more liberal political views, expressed higher levels of trust in their primary care provider, perceived stronger social pressure to vaccinate against COVID-19, and those who received a flu shot during the previous flu season had a stronger intention to vaccinate against COVID-19. Some studies [9–11] show a link between personality and political views, as well as the propensity to succumb to pressure and personality, it is worth investigating the role personality plays in decisions to be vaccinated or otherwise.

While prior studies on non-vaccination predictors have concentrated on specific vaccine beliefs [12, 13], the significance of personality factors in predicting vaccine hesitation has received minimum scholarly attention. Personality has been described as an idiosyncratic style of thinking, feeling, and behaving [14]. Personality encompasses sentiments, attitudes, and perspectives, and is most evident in interpersonal interactions. The Big Five personality traits are well-known for encompassing a broader range of personal characteristics and for investigating the individual effects of personality traits on health behaviours [15]. Extraversion (i.e., being energetic, outgoing, and sociable), agreeableness (i.e., being trustworthy, altruistic, and sympathetic), conscientiousness (i.e., being self-disciplined, dutiful, and thoughtful), emotional stability (i.e., being calm, relaxed, and even-tempered), and openness to experience are the five domains in which personality characteristics can be classified [16, 17].

How people perceive a disease is a crucial factor in understanding the uptake of preventative and health-management interventions such as immunization, particularly for COVID-19. Individuals' cognitive representations or ideas about their condition are referred to as illness perceptions [18]. The self-regulation framework developed by Leventhal and colleagues is the most widely studied conceptual characterization of illness perception [19, 20]. As documented by Broadbent et al. [21], illness perceptions are split into numerous distinct yet interconnected elements. These components have been classified as cognitive or emotional representations of illness [21].

With the onset of the epidemic, several public health promotion goals were critical in the fight against the pandemic [22]. One such strategy was to encourage people to engage in

protective behaviours such as hand washing and physical distancing behaviours which are still vital in this presumably post-pandemic era. According to research, illness perceptions may significantly affect the emotional and behavioural responses to a specific illness [23]. As a result, knowledge of COVID-19 illness perceptions may have important implications for achieving these public health goals of vaccination and devising public health strategies, such as health-promoting programs and their context-specific adaptations.

This study, therefore, aimed to evaluate the role of personality and illness perception on attitudes and hesitancy towards COVID-19 vaccination. Three hypotheses were tested in this study:

H1. Vaccination status will influence attitudes toward COVID-19 vaccination

H2. Sociocultural variables, experience with COVID-19, and psychological variables (anti-vaccine attitudes, personality type and illness belief) will predict vaccine hesitancy

H3. Experience with COVID-19 will moderate the relationship between anti-vaccine attitudes and vaccine hesitancy

## Methods

### Research design

A cross-sectional survey was conducted online using google forms. Participants were recruited via social media platforms-LinkedIn, Twitter, WhatsApp and Facebook. No incentives were provided to social media organisations or participants for taking part in this online survey.

### Sample and sampling technique

Convenience and snowballing sampling techniques were used to collect data for the study. This was done by the researchers' decision to take to reach a sample of the population that was easily available for the study. The researcher recruited assistants to share links to specific age groups (18 and beyond) to get a diverse sample. The forms were set up in a way that participants were required to sign in via email and each account was limited to only one response. Participation was voluntary and participants had access to the questionnaire only after consent was obtained. Participants were also encouraged to share the google link with their contacts. A total of 492 participants were eligible for the study.

### Measures

A structured questionnaire was used to generate data on socio-demographic characteristics, previous experience with COVID-19, personality traits, illness perceptions about COVID-19, and attitude towards the COVID-19 vaccination. The socio-demographic questions included age, sex, and education. Other measures of the questionnaire are as follows.

**Previous experience with COVID-19.** Participants were asked whether they: had had a positive test for COVID-19; are at an increased risk of COVID-19 infection; have had to self-isolate; knew someone with COVID-19 infection, or knew someone who had received a COVID-19 vaccination.

Oxford COVID-19 **Vaccine Hesitancy Scale [24].** This is a seven-item measure based on a survey of 5,114 persons in the United Kingdom. Response options are item-specific and are coded from 1 to 5. A higher score translates to higher vaccine apprehension. The Oxford COVID-19 Vaccine Hesitancy Scale scores are linked to the Vaccine Hesitancy Scale (Shapiro et al. [25]). Items were edited to suit the Ghanaian context.

Vaccine **Attitudes Examination Scale** (VAX) [**26**]**.** This consists of 12 items assessing four factors (vaccine mistrust, future worries, profiteering, and preference for natural immunity). Responses were on a 6-point Likert-type scale ranging from "strongly disagree" to "strongly agree." Higher scores reflect stronger anti-vaccination attitudes.

A **brief version of the Big Five Personality Inventory** (BFI-10) [**27**]**.** This scale was adapted from Rammstedt, B. & John, O. P. [27] the BFI-10 is a 10-item short version of the Big Five Inventory that measures personality across the five main domains and this can take one minute or less to complete.

Brief Illness Perception Questionnaire (IPQ) [**21**]**.** The Brief Illness Perception Questionnaire was developed to assess illness perceptions as specified by the self-regulation approach. The Brief IPQ is a reduced form of the updated illness perception questionnaire for a given disease, consisting of eight items that reflect various aspects of illness perception. The aspects are; consequences, timeline, personal control, treatment control, identity, concern, understanding, and emotional response. Questions were edited to reflect the COVID-19 situation. An example was, "How concerned are you about COVID-19?" Higher ratings indicate a more unfavourable illness perception.

A pre-test of the questionnaires was conducted among 13 people to ensure that the online form is user-friendly.

## Ethics

Ethical approval was sought from the Ethics Committee of the Department of Psychology (DREC/003/21-22), University of Ghana, to ensure that the study follows scientifically approved methods for conducting research. The study was conducted online with a required consent form without which participants could not proceed to respond to the questions.

## Data analysis

Data were analyzed using IBM SPSS Statistics for Windows, version 26 (IBM Corp., Armonk, N.Y., USA). Preliminary analyses were conducted to ensure no violation of the assumptions of normality, linearity, multicollinearity, and homoscedasticity. To determine if sociocultural variables, personality, and other psychological variables predict vaccine hesitancy, hierarchical regression was used to analyze the data. Before performing multivariate analyses, Pearson product-moment correlations, t-tests, and analysis of variance (ANOVA) were employed to explore associations among the predictor variables.

## Results

### Sample

A total of 498 individuals completed or partially completed the survey, but after cleaning the data, only 492 participants were eligible to be used for analysis. The sample comprised participants 18 years and above with a mean age of 25.94 (standard deviation [SD] = 6.52), The majority of the participants in the study were female (55%). There were more vaccinated people in the study (67.5%) than unvaccinated (32.5%). Refer to Table 1 for more information on demographics and Table A in S1 File for the bivariate correlations among the predictor variables.

### Influence of vaccination status on attitudes toward COVID-19 vaccination

A one-way between-groups multivariate analysis of variance was performed to investigate how vaccine status influences attitudes towards vaccination. Vaccine mistrust, future effects,

**Table 1. Frequency distribution of demographical characteristics (N = 492).**

| Variables | | Frequency | Percentages |
|---|---|---|---|
| Age (18–68) | M(25.94), S.D(6.52) | 492 | 100 |
| Gender | | | |
| | Male | 227 | 46.1 |
| | Female | 265 | 53.9 |
| Marital Status | | | |
| | Single | 426 | 86.6 |
| | Married | 61 | 12.4 |
| | Separated/Divorced | 5 | 1.0 |
| Religion | | | |
| | Christian | 458 | 93.1 |
| | Muslim | 23 | 4.7 |
| | Others | 11 | 2.2 |
| Educational Level | | | |
| | ≤Secondary | 19 | 3.9 |
| | Tertiary | 473 | 96.1 |
| Vaccination Status | | | |
| | Vaccinated | 332 | 67.5 |
| | Unvaccinated | 160 | 32.5 |
| Employment Status | | | |
| | Self-employed/Informal | 70 | 14.2 |
| | Formal | 259 | 52.6 |
| | Unemployed | 163 | 33.2 |
| Ethnic background | | | |
| | Akan | 247 | 50.2 |
| | Ga/Ga-Adangbe | 61 | 12.4 |
| | Ewe | 83 | 16.9 |
| | Others | 100 | 20.3 |
| | Missing | 1 | .2 |

profiteering, and natural immunity were employed as dependent variables. On the combined dependent variables, there was a statistically significant difference between the vaccinated and unvaccinated, $F_{(4, 492)} = 27.36$, $p = 0.000$; Wilks' Lambda = .82; partial eta squared = .18. Using a Bonferroni corrected alpha level of .013, all the variables remained statistically significant when the results for the dependent variables were analysed separately. A closer examination of the means in Table 2 shows that the unvaccinated have greater mistrust for vaccines (M = 9.86, SD = 3.14) when compared to their vaccinated cohort at p = .000. Regarding the belief that vaccines may have some effects in future, the unvaccinated recorded higher means

**Table 2. Summary of MANOVA results for anti-vaccine attitudes measures.**

| Variable | Vaccine Status | | | | |
|---|---|---|---|---|---|
| | Vaccinated(332) | Unvaccinated(160) | | | |
| | Mean (SD) | Mean(SD) | F | p | $\eta^2$ |
| Vaccine Mistrust | 7.1±2.56 | 9.86±3.14 | 107.27 | .000 | 0.18 |
| Future Effects | 11.09±2.12 | 11.90±2.49 | 13.91 | .000 | 0.03 |
| Profiteering | 8.71±2.64 | 10.01±2.75 | 26.52 | .000 | 0.05 |
| Natural Immunity | 11.80±3.96 | 11.88±3.98 | 18.73 | .000 | 0.04 |

(M = 11.90, SD = 2.49) compared to the vaccinated (M = 11.09, SD = 2.12). Also, those who have not been vaccinated believe that vaccines are largely for-profit (M = 10.01, SD = 2.75) compared to those vaccinated (M = 8.71, SD = 2.64). Those unvaccinated also hold the view that natural immunity is better than vaccines (M = 11.88, SD = 3.98) compared to those who have been vaccinated (M = 11.80, SD = 3.96).

## Sociocultural variables, experience with COVID-19, and psychological variables (anti-vaccine attitudes, personality type and illness belief) will predict vaccine hesitancy

Hierarchical multiple regression was used to assess this hypothesis. This method of regression was used to determine how the various independent variables grouped as models show a significant improvement in the $R^2$ (i.e., the proportion of the variance explained by the model). The basic assumptions of regression normality, linearity, homoscedasticity, and multicollinearity were tested and these assumptions were met without any significant violations. According to Pallant [28] multicollinearity in regression occurs when the independent variables are highly correlated (r = .7 and above). All of the variables (subscales) as used in the regression analysis correlated below .7 as seen in Table A in S1 File. Also, if the standard errors are small and the precision for the variables of interest is high, multicollinearity can be waived. Sociocultural variables were entered in Step 1, explaining 6% of the variance in Vaccine hesitancy. Being single was the only variable with a beta value reaching statistical significance in the first model (beta = −.33, $p < .05$). In model 2, when the big first personality factors openness, conscientiousness, extraversion, agreeableness, and neuroticism were entered, none was statistically significant in predicting vaccine hesitancy. Experience with COVID-19 was entered in model 3 and was significant (beta = .45, $p < .01$). In model 4, components of illness belief were entered leading to an $R^2$ change of .14 and this model explained 38% of the variance. In model 5, subscales of anti-vaccine attitudes which are mistrust of vaccine, future effects, profiteering and natural immunity were entered with an $R^2$ change of .23 and the final model explains 60% of the variance. R squared change = .60, F change (4, 492) = 63.84, $p < .001$. No sociocultural variable significantly predicted vaccine hesitancy in model 5, extraversion (beta = −.09, $p < .001$), experience with COVID-19 (beta = .23, $p < .001$), identity (beta = .07, $p < .05$), illness concern (beta = −.11, p < .001), emotional representation (beta = −.08, $p < .001$), treatment control (beta = .07, $p < .001$), concern about future effects (beta = .09, $p < .001$), profiteering (beta = .09, $p < .05$) and vaccine mistrust (beta = .49, $p < .001$) were significant with vaccine mistrust having the highest beta value (Table 3).

## Experience with COVID-19 will moderate the relationship between anti-vaccine attitudes and vaccine hesitancy

To test the hypothesis that experience with COVID-19 will moderate the relationship between Vaccine hesitancy and anti-vaccine attitudes, simple moderation analysis was performed using Model 1 of the PROCESS macro version 4.0 with a confidence interval of 95% and bootstraps set at 5,000. The variables accounted for a significant amount of variance in vaccine hesitancy, $R^2 = .43$, $F(3,488) = 181.6548$, $p < 0.001$. As seen in Table 4, the interaction between vaccine attitude and vaccine hesitancy was found to be statistically significant {$b = .08$, 95% C. I (.0333, .1227), $p < 0.01$}, indicating that the moderating effect was significant. Thus, the hypothesis was supported.

Each of the simple slope tests revealed a significant positive association between predictor and outcome variables at low moderation {Conditional effect = .3648, 95% C.I (.2791, .4705), $p < 0.01$}; at middle moderation, {Conditional effect = .5309, 99% C.I (.4654, .5963), $p < 0.01$}

**Table 3. Summary of regression analysis.**

| Variable | Model 1 | | | Model 2 | | | Model 3 | | | Model 4 | | | Model 5 | | |
|---|---|---|---|---|---|---|---|---|---|---|---|---|---|---|---|
| | B | SE B | β | B | SE B | β | B | SE B | β | B | SE B | β | B | SE B | β |
| **Sociocultural Variables** | | | | | | | | | | | | | | | |
| Gender = Female | .19 | .71 | .01 | .40 | .72 | .03 | .37 | .66 | .02 | .68 | .61 | .04 | .13 | .50 | .01 |
| Ages < = 23 (vs 31+) | .94 | 1.48 | .06 | .87 | 1.50 | .06 | .28 | 1.36 | .02 | -.56 | 1.25 | -.04 | -.81 | -4.00 | -.05 |
| Ages 24–30 (vs 31+) | -.58 | 1.35 | -.04 | -.64 | 1.36 | -.04 | -.32 | 1.24 | -.02 | -.42 | 1.13 | -.03 | -.54 | .91 | -.04 |
| Akan (vs other ethnicities) | -.52 | .00 | -.03 | -.46 | -2.00 | -.03 | -.23 | .91 | -.02 | -.43 | .83 | -.03 | -.60 | .67 | -.04 |
| Ga-Adangbe (vs other ethnicities) | 1.25 | 1.33 | .05 | 1.21 | 1.33 | .05 | .75 | 1.21 | .03 | .19 | 1.11 | .01 | .62 | .89 | .03 |
| Ewe (vs other ethnicities) | 1.69 | 1.26 | .08 | 1.55 | 1.26 | .08 | 1.46 | 1.15 | .07 | 1.16 | 1.05 | .06 | .85 | .85 | .04 |
| Single (vs. widowed/divorced) | -7.37 | 3.67 | **-.33**\* | -7.64 | 3.68 | **-.34**\* | -7.59 | 3.35 | **-.34**\* | -5.94 | 3.09 | -.26 | -1.91 | 2.51 | -.09 |
| Married (vs. widowed/divorced) | -6.77 | 3.71 | -.29 | -7.06 | 3.72 | -.30 | -6.71 | 3.39 | **-.29**\* | -5.55 | 3.10 | -.24 | -1.57 | 2.52 | -.07 |
| Tertiary (vs < = secondary) | -3.04 | 1.89 | -.08 | -3.24 | 1.90 | -.08 | -4.11 | 1.73 | **-.10**\* | -3.94 | 1.58 | -.10 | -1.84 | 1.28 | -.05 |
| **Variable** | **Model 1** | | | **Model 2** | | | **Model 3** | | | **Model 4** | | | **Model 5** | | |
| | B | SE B | β | B | SE B | β | B | SE B | β | B | SE B | β | B | SE B | β |
| Unemployed (vs. self-employed) | -.33 | 1.19 | -.02 | -.29 | 1.20 | -.02 | .43 | 1.10 | .03 | .76 | 1.02 | .05 | .53 | .82 | .03 |
| Formally employed (vs. self-employed) | -1.32 | 1.08 | -.09 | -1.23 | 1.09 | -.08 | .04 | .99 | .00 | .52 | .91 | .03 | .31 | .74 | .02 |
| Christian (vs others) | -.46 | 2.41 | -.02 | -1.19 | 2.43 | -.04 | -2.99 | 2.22 | -.10 | -1.52 | 2.06 | -.05 | -1.21 | 1.67 | -.04 |
| Muslim (vs others) | -1.85 | 3.01 | -.05 | -2.55 | 3.04 | -.07 | -4.18 | 2.77 | -.12 | -2.27 | 2.58 | -.06 | -3.03 | 2.08 | -.08 |
| Greater Accra (vs Central Region) | -2.08 | 1.63 | -.13 | -1.93 | 1.64 | -.12 | -.92 | 1.50 | -.06 | -1.13 | 1.37 | -.07 | .46 | 1.11 | .03 |
| 4 Northern regions (vs Central Region) | .08 | 2.12 | .00 | .22 | 2.12 | .01 | .13 | 1.93 | .00 | .11 | 1.77 | .00 | 1.79 | 1.43 | .06 |
| Ashanti (vs Central Region) | -.68 | 1.96 | -.03 | -.70 | 1.97 | -.03 | -1.45 | 1.79 | -.06 | -1.55 | 1.64 | -.06 | -.18 | 1.33 | -.01 |
| Western (vs Central Region) | .00 | 2.31 | .00 | .27 | 2.32 | .01 | .70 | 2.11 | .02 | -1.22 | 1.93 | -.03 | .00 | 1.56 | .00 |
| Volta (vs Central Region) | -2.52 | 2.59 | -.06 | -2.12 | 2.62 | -.05 | -.99 | 2.39 | -.02 | -1.18 | 2.18 | -.03 | .28 | 1.76 | .01 |
| Eastern (vs Central Region) | -.34 | 2.29 | -.01 | -.35 | 2.31 | -.01 | -.20 | 2.11 | -.01 | -.39 | 1.92 | -.01 | 1.02 | 1.55 | .03 |
| **Psychological variables** | | | | | | | | | | | | | | | |
| Openness | | | | .02 | .24 | .01 | .06 | .22 | .01 | .01 | .20 | .00 | -.02 | .16 | .00 |
| **Variable** | **Model 1** | | | **Model 2** | | | **Model 3** | | | **Model 4** | | | **Model 5** | | |
| | B | SE B | β | B | SE B | β | B | SE B | β | B | SE B | β | B | SE B | β |
| Extraversion | | | | -.19 | .19 | -.05 | -.32 | .18 | -.08 | -.36 | .16 | -.09 | -.35 | .13 | **-.09**\*\* |
| Agreeableness | | | | .46 | .30 | .08 | -.51 | .29 | -.08 | -.36 | .27 | -.06 | -.39 | .22 | -.06 |
| neuroticism | | | | -.20 | .22 | -.04 | -.25 | .20 | -.06 | -.08 | .19 | -.02 | -.04 | .15 | -.01 |
| Experience with COVID-19 | | | | | | | 2.83 | .29 | **.45**\*\* | 2.18 | .28 | **.35**\*\* | 1.45 | .23 | **.23**\*\* |
| Consequences | | | | | | | | | | -.13 | .12 | -.04 | -.03 | .10 | -.01 |
| Timeline | | | | | | | | | | -.19 | .12 | -.07 | -.12 | .10 | -.04 |
| Identity | | | | | | | | | | .34 | .13 | **.11**\*\* | .23 | .11 | **.07**\* |
| Illness Concern | | | | | | | | | | -.66 | .13 | **-.24**\*\* | -.30 | .11 | **-.11**\*\* |
| Emotional representation | | | | | | | | | | -.05 | .13 | -.02 | -.23 | .11 | **-.08**\* |
| Personal Control | | | | | | | | | | -.27 | .12 | **-.09**\* | -.11 | .10 | -.04 |
| Treatment Control | | | | | | | | | | .75 | .15 | **.22**\*\* | .24 | .12 | **.07**\* |
| Coherence | | | | | | | | | | -.07 | .15 | -.02 | .01 | .12 | .00 |
| Vaccine Mistrust | | | | | | | | | | | | | 1.22 | .10 | **.49**\*\* |
| Future effects | | | | | | | | | | | | | .31 | .12 | **.09**\*\* |
| Profiteering | | | | | | | | | | | | | .26 | .12 | **.09**\* |
| Natural immunity | | | | | | | | | | | | | -.10 | .12 | -.03 |
| $R^2$ | | .06 | | | .07 | | | .23 | | | .38 | | | .60 | |
| $R^2$ change | | | | | .01 | | | .16 | | | .14 | | | .23 | |

NB. B- unstandardized Beta; SE, standard error; β-Standardized Beta.

\*p< .05,

\*\* p < .001.

The figures in bold are statistically significant.

**Table 4. Summary of the results of the moderation effect.**

| Model | B | SE | t | P |
|---|---|---|---|---|
| Constant | 16.91 | .29 | 59.13 | .001 |
| Anti-vaccination Attitudes | .48 | .03 | 14.93 | .001 |
| Experience | 1.79 | .21 | 8.38 | .001 |
| Int_1 | .08 | .02 | 3.43 | .001 |

B = coefficient/ slope of the intercept; SE = standard error; p = significant level; Int_1 = interaction.

and at high moderation, {Conditional effect = .6089, 99% C.I (.5203, .6974), $p<0.01$}. The results identified experience with COVID-19 as a positive moderator of the relationship between anti-vaccine attitudes and vaccine hesitancy (Fig 1).

The graph (Fig 1) shows how the relationship between the predictor and outcome variables change according to the level of the moderator. Individuals with high levels of COVID-19 experience have low vaccine hesitancy when anti-vaccination attitudes are low. For participants with low level of COVID-19 experience, even with low anti-vaccination attitudes, they have substantially higher levels of vaccine hesitancy.

## Discussion

This study examined the factors that influenced vaccine uptake by the general public. In this study, we observed that sociocultural characteristics did not predict vaccine hesitancy, extraversion, on the other hand, was the only significant psychological variable that predicted vaccine hesitancy. In addition, illness beliefs about COVID-19 influenced anti-vaccine attitudes, and experience with COVID-19 and attitudes towards vaccination influenced vaccine hesitancy.

To begin with, vaccination has been a topical issue in Ghana, as well as in other countries. People are required to be vaccinated before flying in and out of the country. Some countries required people to be vaccinated before entering certain places and facilities. In this study, 67.5% had been vaccinated and negative attitudes towards vaccination were higher in the unvaccinated. Interestingly, however, the mean difference between the vaccinated and unvaccinated on vaccine mistrust was close, thereby indicating that although people may have been vaccinated, they also exhibit some level of mistrust towards the vaccines, and this could be a result of the speculations around COVID-19 vaccines. The majority of the participants (59.7%) believed that the vaccines were safe, yet, responded to being sceptical about their efficacy.

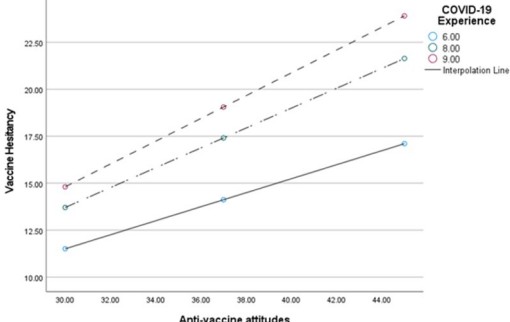

**Fig 1. Interaction effect between vaccine hesitancy and anti-vaccine attitudes as moderated by experience with COVID-19.** NB: COVID-19 Experience = Moderator. '6.00' = Low COVID-19 Experience; '8.00' = Mid Covid-19 Experience; '9.00' = High Covid-19 Experience.

The researchers in this study posited that personality type would greatly influence vaccine hesitancy but per our regression analysis, that was not found. Only extraversion was found to be a significant predictor. This is concordant with a finding by Halstead et al. [29] who also found extraversion as a predictor of vaccine hesitancy. However, the bivariate correlation showed a positive association between conscientiousness and anti-vaccine attitudes. Conscientiousness as a personality type is positively connected with behaviours such as cautiousness, orderliness, and dutifulness [30, 31] which may invariably deter people from receiving the vaccine or have negative attitudes toward it considering the swift rollout of COVID-19 vaccines as compared to other vaccines. Vaccines often take 10–15 years to develop, however, the current COVID-19 vaccine was developed in less than three years [32]. Some vaccines require a single dose, whilst others recommend double doses and still require booster shots. The argument concerning the ability to pass on the virus or suffer mild symptoms even after vaccination has been rife since the vaccines were produced and distributed globally. Conscientiousness is largely associated with engaging in an activity thoroughly and dutifully. The unsteady nature of the number of doses and COVID-19 vaccine efficacy as compared to other vaccines may inform conscientious people in their attitudes towards the COVID-19 vaccines. Other studies have expressed mixed findings regarding the role personality plays in vaccine hesitancy [33, 34] indicating that there are inconsistent findings regarding personality and vaccine acceptance.

In this study, we found out that COVID-19 experience results in a unit increase in vaccine hesitancy. This could be explained by the fact that the vast majority of respondents (85.6%) reported they had never been infected with COVID-19 at the time the study was conducted, and 52.6% also had people close to them who had tested positive for COVID-19. A huge majority of 96.5% of the sample had people close to them who had been vaccinated. Vaccine hesitancy could be a result of the belief in vaccination by proxy. People will potentially be hesitant to get vaccinated if they have never been formally diagnosed as contracting the virus despite other people around them testing positive. Also, considering that almost the entire sample had people close to them who had been vaccinated, this could result in a lax disposition and not create a sense of urgency since they might feel safe and not exposed to any clear and present danger because their close contacts have been vaccinated. The study also found the experience with COVID-19 to moderate the relationship between anti-vaccine attitudes and vaccine hesitancy. As seen from the moderation, as anti-vaccine attitudes increase, people with lower experience still had lower vaccine hesitancy but those with high COVID-19 experience had higher vaccine hesitancy (Fig 1). A reason for this could be that those who may have tested positive for COVID-19 before may perceive themselves to have been naturally inoculated, hence, making them less keen to be vaccinated.

One key variable in this study worth noting is illness belief. Some of the dimensions of illness belief (identity, concern, emotional representation, and treatment control) significantly influenced vaccine hesitancy in the regression analysis. Concern about COVID-19 infections had a negative beta value meaning that a unit decrease in concern about COVID-19 led to an increase in vaccine hesitancy. Treatment control refers to how a person thinks treatment can cure COVID-19 disease. As expected, when one does not view the infection as dangerous but easily treatable with medication, the individual will be less likely to want to be vaccinated. The emotional representation of COVID-19 focuses on fear, anger and distresses endured as a result of the illness. The variable had a negative beta value meaning that participants who expressed less fear, anger, distress and other negative emotions about COVID-19 were more hesitant to get the vaccine because the disease did not appear to affect them adversely.

Anti-vaccine attitudes (bar natural immunity) proved to predict vaccine hesitancy and having great concern about the future effects of the vaccine led to more vaccine hesitancy. As

one's trust in the vaccines goes down, vaccine hesitancy increases. This observation is supported by previous research which found that concerns about future unforeseen side effects were the most important determinants of both uncertainty and unwillingness to vaccinate against COVID-19 [35–37].

The study however acknowledges the main limitation of online data collection approaches where only participants who were literate and had access to the internet could participate in the study. Similarly, the majority of participants were female with tertiary-level education and this has also been previously reported as a limitation of online studies [38]. Another limitation was that the potential bias in socio-demographic background places a limit on the generalizability of the study. Again, there is the possibility that knowledge of COVID-19 vaccinations may have led to the provision of socially desirable answers. Despite these limitations, this study's findings highlight some findings with public health and policy implications. With many countries lifting several restrictions due to the pandemic, it is increasingly becoming necessary to address concerns about hesitancy and improve vaccine uptake since the COVID-19 infection continues to persist.

## Conclusion

The current study found that vaccination attitudes and illness beliefs about COVID-19 influenced vaccine hesitancy. The outcomes from this study have implications for achieving public health goals and developing strategies for reaching optimal vaccination targets and attaining herd immunity when enough people have been vaccinated against COVID-19. Vaccine promotion messages and activities should highlight the benefit-risk balance of the vaccines while providing reassurance to the public.

## Supporting information

**S1 File.** Table A. Inter-correlations between Illness beliefs, Big five personality types, components of vaccine attitudes, combined anti-vaccine attitudes and vaccine hesitancy. Table B. Summary of hierarchical regression results.
(DOCX)

## Acknowledgments

The authors would like to thank all participants for their responses to this research.

## Author Contributions

**Conceptualization:** Eric Nanteer-Oteng.

**Data curation:** Eric Nanteer-Oteng.

**Formal analysis:** Eric Nanteer-Oteng.

**Investigation:** Irene A. Kretchy.

**Methodology:** Eric Nanteer-Oteng, Irene A. Kretchy.

**Supervision:** Irene A. Kretchy, Joseph Osafo.

**Validation:** Irene A. Kretchy.

**Visualization:** Eric Nanteer-Oteng, Irene A. Kretchy.

**Writing – original draft:** Eric Nanteer-Oteng.

**Writing – review & editing:** Eric Nanteer-Oteng, Irene A. Kretchy, Deborah Odum Nanteer, James-Paul Kretchy, Joseph Osafo.

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
