## [Decision Letter · Decision Letter 0]

30 Aug 2022

PGPH-D-22-00710

Hesitancy towards COVID-19 vaccination: The role of personality traits, anti-vaccine attitudes and illness perception

Dear Dr. Eric Nanteer-Oteng,

Thank you for submitting your manuscript to PLOS Global Public Health. After careful consideration, we feel that it has merit but does not fully meet PLOS Global Public Health’s publication criteria as it currently stands. Therefore, we invite you to submit a revised version of the manuscript that addresses the points raised during the review process.

We look forward to receiving your revised manuscript.

Kind regards,

Ethel Leonor Noia Maciel, Ph.D.

Academic Editor

Journal Requirements:

2. Please amend your Data Availability Statement and indicate where the data may be found.

Additional Editor Comments (if provided):

The manuscript titled "Hesitancy Regarding COVID-19 Vaccination: The Role of Personality Traits, Anti-Vaccine Attitudes, and Disease Perception" is a topic of great interest now, as the COVID-19 pandemic has spread anti-vaccine movements across the world. world, impacting the vaccination coverage of all other vaccines, which requires even more efforts to defend vaccination.

I send the evaluations of reviewers 1 and 2 in addition to suggesting that alignment be made in the introduction with greater depth to answer what the MS adds to the field of knowledge.

Also in the section methods, a more detailed description of the sample will help readers better understand the limitations of the study.

Reviewers' comments:

Reviewer's Responses to Questions

**Comments to the Author**

1. Does this manuscript meet PLOS Global Public Health’s publication criteria? Is the manuscript technically sound, and do the data support the conclusions? The manuscript must describe methodologically and ethically rigorous research with conclusions that are appropriately drawn based on the data presented.

Reviewer #1: Partly

Reviewer #2: Yes

2. Has the statistical analysis been performed appropriately and rigorously?

Reviewer #1: I don't know

Reviewer #2: Yes

3. Have the authors made all data underlying the findings in their manuscript fully available (please refer to the Data Availability Statement at the start of the manuscript PDF file)?

Reviewer #1: Yes

Reviewer #2: Yes

4. Is the manuscript presented in an intelligible fashion and written in standard English?

Reviewer #1: No

Reviewer #2: Yes

5. Review Comments to the Author

Reviewer #1: Thank you so much for submitting the manuscript titled "Hesitancy towards COVID-19 vaccination: The role of personality traits, anti-vaccine attitudes and illness perception". Such topic is of great interest now as COVID-19 is entering a new era of being endemic rather than pandemic which necessitate more vaccination advocacy efforts. Additionally, articulating the hypotheses in all the manuscript sections makes the conclusion very clear and logic. However, i find it important to address the following points before this manuscript is being considered for publishing:

Generally, the manuscript needs more improvement in the formatting and mild English professional revision.

Introduction:

1- This is a just reviewer friendly request and not substantial issue in the manuscript itself: there are no page numbers nor line numbers in the manuscript which makes it difficult to point out specific improvements with examples.

2- The introduction is being too broad. It could be shorten and be more focused on COVID-19 vaccine and the gaps that this study would cover.

Methods:

1- It is important to elaborate on the sampling method and it would be interesting to know how limiting one answer to one person was achieved using Google forms.

Results:

1- Please consider re-formating tables (including the ones in the appendices) as they are not reader-friendly and not well -organized. For example, Table.1: Different categories like: vaccination status, Employment status. etc need to be labeled. Also no need to keep space and add many dashes for Mean and SD while they do not exist for many variables. Better to keep mean and SD under age variable and remove the extra columns.

2- It is a good idea to list the hypotheses and describe the results and analysis in line with them. I would suggest to put the associated table with each hypothesis in the result section. For example, hierarchal linear regression table could be added to H2 in the results instead of being an appendix.

3- In table 2, it is not clear "statistically" how such a mild difference between vaccinated and unvaccinated groups would be highly significant. It would be beneficial to list the number of sample in each category to better demonstrate the result.

3- It would be beneficial for the reader to get information on the level of multicollinearity particularly with with variables like vaccine attitude and COVID-19 vaccine hesitancy. And such results should be explained and justified through the hierarchy linear regression.

4- The tables labels are not being referred to properly in the result descriptions which is confusing (table no...)

5- It would be great if the selection of predictors for each model in the hierarchy linear regression is being explained.

6- In the moderation effect simple slope analysis; what defines high, moderate or low COVID-19 vaccine hesitancy level? Are they subjective cut-offs? As explained in the scale used, the scores indicate comparativeness rather than categorization or leveling.

Discussion:

The results were explained well in the discussion. The only point is on the limitations: two important limitations were not considered in your description:

1- The results of this study cannot be generalized due to the bias in participants socio-demographic

background.

2- Level of participants' education and consequently COVID-19 and COVID-19 vaccination knowledge might

have an influence on the reporting toward more socially desirable answers.

socially desirable answers.

Reviewer #2: I thought the work was well executed, on a topic of great importance for the current moment. The methods and statistical analyzes used are adequate. The way of accessing the sample, the work database, was a possible choice that made it possible to obtain the data. The electronic means of procurement is a valuable possibility. There are, however, some precautions to be taken so that the sample, obtained via online, is random and representative of the population of interest. I believe that this is the only criticism of the work, but it does not diminish its merit, considering that the authors took some care in this regard.

6. PLOS authors have the option to publish the peer review history of their article (what does this mean?). If published, this will include your full peer review and any attached files.

**Do you want your identity to be public for this peer review?** For information about this choice, including consent withdrawal, please see our Privacy Policy.

Reviewer #1: No

Reviewer #2: **Yes: **ADELMO INACIO BERTOLDE

---

## [Decision Letter · Decision Letter 1]

31 Oct 2022

PGPH-D-22-00710R1

Hesitancy towards COVID-19 vaccination: The role of personality traits, anti-vaccine attitudes and illness perception

Dear Dr. Nanteer-Oteng,

Thank you for submitting your manuscript to PLOS Global Public Health. After careful consideration, we feel that it has merit but does not fully meet PLOS Global Public Health’s publication criteria as it currently stands. Therefore, we invite you to submit a revised version of the manuscript that addresses the points raised during the review process.

We look forward to receiving your revised manuscript.

Kind regards,

Abram L. Wagner, PhD, MPH

Academic Editor

Journal Requirements:

2. Please provide separate figure files in .tif or .eps format only and remove any figures embedded in your manuscript file. Please also ensure that all files are under our size limit of 10MB.

Additional Editor Comments (if provided):

Reviewers' comments:

Reviewer's Responses to Questions

**Comments to the Author**

1. If the authors have adequately addressed your comments raised in a previous round of review and you feel that this manuscript is now acceptable for publication, you may indicate that here to bypass the “Comments to the Author” section, enter your conflict of interest statement in the “Confidential to Editor” section, and submit your "Accept" recommendation.

Reviewer #1: (No Response)

2. Does this manuscript meet PLOS Global Public Health’s publication criteria? Is the manuscript technically sound, and do the data support the conclusions? The manuscript must describe methodologically and ethically rigorous research with conclusions that are appropriately drawn based on the data presented.

Reviewer #1: Yes

3. Has the statistical analysis been performed appropriately and rigorously?

Reviewer #1: Yes

4. Have the authors made all data underlying the findings in their manuscript fully available (please refer to the Data Availability Statement at the start of the manuscript PDF file)?

Reviewer #1: Yes

5. Is the manuscript presented in an intelligible fashion and written in standard English?

Reviewer #1: No

6. Review Comments to the Author

Reviewer #1: The new submission has addressed a lot of the comments of editors which is great. I have few points left for the authors to consider:

1- The multicollinearity between COVID-vaccination hesitancy and vaccine attitude is still un explain in the manuscript. These two factors were input in one model which might jeopardize the result. Please refer to S1 (appendix)

2- The manuscript still needs revision for typos.

3- No need to add "hypothesis..." to the subtitles and tables. The current structure helps the reader to understand that it is the hypothesis.

7. PLOS authors have the option to publish the peer review history of their article (what does this mean?). If published, this will include your full peer review and any attached files.

**Do you want your identity to be public for this peer review?** For information about this choice, including consent withdrawal, please see our Privacy Policy.

Reviewer #1: No

---

## [Editor Report · Decision Letter 2]

18 Nov 2022

PGPH-D-22-00710R2

Hesitancy towards COVID-19 vaccination: The role of personality traits, anti-vaccine attitudes and illness perception

Dear Dr. Nanteer-Oteng,

Thank you for submitting your manuscript to PLOS Global Public Health. After careful consideration, we feel that it has merit but does not fully meet PLOS Global Public Health’s publication criteria as it currently stands. Therefore, we invite you to submit a revised version of the manuscript that addresses the points raised during the review process.

Overall I just have a min9or comment about Figure 1 and can process this manuscript quickly after receiving that edit.

We look forward to receiving your revised manuscript.

Kind regards,

Abram L. Wagner, PhD, MPH

Academic Editor

Journal Requirements:

Additional Editor Comments (if provided):

Can you write a more detailed caption for Figure 1 and include more of a description on the legend? As is I'm not sure what the lines mean.
---

## [Editor Report · Decision Letter 3]

6 Dec 2022

Hesitancy towards COVID-19 vaccination: The role of personality traits, anti-vaccine attitudes and illness perception

PGPH-D-22-00710R3

Dear Mr Nanteer-Oteng,

We are pleased to inform you that your manuscript 'Hesitancy towards COVID-19 vaccination: The role of personality traits, anti-vaccine attitudes and illness perception' has been provisionally accepted for publication in PLOS Global Public Health.

Best regards,

Abram L. Wagner, PhD, MPH

Academic Editor